# Enhanced Antimicrobial Cellulose/Chitosan/ZnO Biodegradable Composite Membrane

**DOI:** 10.3390/membranes12020239

**Published:** 2022-02-18

**Authors:** Xiaolong Sun, Longfei Yin, Huayue Zhu, Junhao Zhu, Jiahuan Hu, Xi Luo, He Huang, Yongqian Fu

**Affiliations:** 1State Key Laboratory of Material-Oriented Chemical Engineering, School of Pharmaceutical Sciences, Nanjing Tech University, No. 30 Puzhu Road, Nanjing 211816, China; sunxiaolong198901@163.com (X.S.); huangh@njtech.edu.cn (H.H.); 2Institute of Biomass Resources, Taizhou University, Taizhou 318000, China; lfyin@tzc.edu.cn (L.Y.); zhuhuayue@126.com (H.Z.); hu1366257348@163.com (J.H.); tzcluoxi@163.com (X.L.); 3Zhejiang Kingsun Eco-Pack Co., Ltd., Taizhou 317000, China; 13857709226@163.com

**Keywords:** composite membrane, phenyllactic acid, antimicrobial activity

## Abstract

In this study, chitosan and sugarcane cellulose were used as film-forming materials, while the inorganic agent zinc oxide (ZnO) and natural compound phenyllactic acid (PA) were used as the main bacteriostatic components to fabricate biodegradable antimicrobial composite membranes. The water absorption and antimicrobial properties were investigated by adjusting the concentration of PA. The scanning electron microscopy (SEM), Fourier transform infrared spectroscopy (FTIR) and X-ray diffraction (XRD) results demonstrated that the components of the composite membrane were successfully integrated. The addition of ZnO improved the mechanical and antimicrobial properties of the composite membrane, while the addition of PA with high crystallinity significantly reduced the water absorption and swelling. Moreover, the addition of 0.5% PA greatly improved the water absorption of the composite membrane. The results of antimicrobial experiments showed that PA improved the antimicrobial activity of the composite membrane against *Staphylococcus aureus*, *Escherichia coli*, *Aspergillus niger* and *Penicillium rubens*. Among them, 0.3% PA had the best antimicrobial effect against *S. aureus*, *E. coli* and *A. niger*, while 0.7% PA had the best antimicrobial effect against *P. rubens*.

## 1. Introduction

Since their invention, plastic products have been widely used because of their convenience and low price, which has resulted in a large amount of plastic waste [1]. However, even with single-use packaging, the incidence of foodborne diseases remains high, necessitating approaches to inhibit the growth and reproduction of microorganisms [2]. The addition of antibacterial compounds to biodegradable materials can not only reduce plastic pollution, but delay the spoilage of food and reduce the occurrence of foodborne diseases [3]. Such biodegradable antimicrobial materials can be widely used in many fields such as the food industry [4], textile industry [5], and biomedicine [6]. Cellulose membrane materials are characterized by high strength, high transparency, and air-tightness [7,8,9,10]. However, they do not possess antimicrobial properties [11]. Chitosan, a natural product obtained by partial or complete deacetylation of chitin, has certain antimicrobial properties [12,13], biocompatibility [14], film-forming degradability [15], and anticoagulant effect [16], which makes it a promising dressing material in the biomedical field [6]. However, membrane materials prepared from pure chitosan have disadvantages such as low strength and brittleness [17], and its independent applications are limited. The preparation of composite membranes by blending cellulose with chitosan compensates for the deficiencies of both components to some extent. Strong hydrogen bonds between cellulose and chitosan form the basis for the construction of composite membranes with good strength, biocompatibility, and antimicrobial activity [18]. However, there are still some challenges in the development of chitosan/cellulose composite membrane materials. Chitosan in composite membranes has limited and unstable antimicrobial activity, and abiotic factors such as molecular weight, degree of deacetylation, crystallinity, and pH can affect it as a mixture, which varies greatly depending on the source of raw materials and production methods, and it can even promote the growth of bacteria rather than inhibit them under certain conditions [19]. It has also been shown that a low chitosan concentration on the surface of composite membranes might lead to their poor inhibitory capacity. Increasing the addition of chitosan can improve the inhibitory capacity but reduces the oxygen barrier properties [20].

Various strategies have been investigated to overcome these defects in composite membranes. For example, antimicrobial properties of composite membranes were improved by modification or combination with antimicrobial agents. According to the research of Pirsa et al., the addition of inhibitors such as TiO_2_, polyaniline, SiO_2_, KMnO_4_ and MgO to biodegradable materials improved their antimicrobial [21,22,23,24] and physicochemical [25,26,27,28] properties. Phenyllactic acid (PA) is a natural antibacterial compound that is stable to acid and heat, soluble in water, and its physicochemical properties are stable over a wide pH range [29]. PA has a broad-spectrum inhibitory effect on a variety of Gram-positive bacteria, Gram-negative bacteria, and fungi, and its site of action is mainly thought to be located in the cell wall, similar to the mechanism of action of lysozyme [30]. Grafting of PA onto chitosan significantly increased its antimicrobial activity, which was positively correlated with the grafting rate of chitosan derivatives and also introduced antioxidant properties [31]. However, the cost of PA as an antimicrobial material is high, and a synergistic antimicrobial strategy needs to be developed to maintain high antimicrobial activity while reducing the amount of PA. Zinc oxide (ZnO) has been widely used as an inorganic bacterial inhibitor with slow-release, long-lasting, broad-spectrum antibacterial properties, and thermal stability [32]. It is widely believed to have three inhibition mechanisms: contact adsorption, metal ion solubilization, and photocatalytic production of reactive oxygen species [32,33,34,35,36,37,38]. Furthermore, ZnO nanoparticles can provide a larger specific surface area, increase the contact with bacteria and provide more active sites for the photocatalytic process [39,40,41], which can further enhance the antibacterial effect. The use of chitosan/cellulose hybrid membranes as carrier materials can provide a new form of bacterial inhibition with a synergistic effect. As reported in previous studies, adding ZnO with strong oxidizing power can effectively improve the antibacterial activity of composite membranes [42]. In packaging materials, the addition of nano-ZnO to chitosan composite membranes not only effectively reduced the fruit decay rate, but improved the degradation rate of residual pesticides and provided better mechanical properties than pure chitosan membranes [43].

In this study, cellulose and chitosan were used as the main body of the biodegradable membranes, while ZnO nanoparticles were dispersed into the membranes together with PA to synergistically enhance its antimicrobial properties. The mechanical and antimicrobial properties of the composite membranes were improved by ZnO, while water absorption, swelling, and antimicrobial properties can be tuned by changing the ratio of PA. The biodegradable antimicrobial membranes prepared in this study are biocompatible, while the combination of ZnO and PA can improve the mechanical properties of the composite membranes and enhance their antimicrobial activity against bacteria and fungi. These membranes are expected to be applied in new antimicrobial food packaging and biomedical applications.

## 2. Materials and Methods

### 2.1. Materials and Strains

Sugarcane fibers were prepared in our laboratory. Chitosan (90–95%, 50–800 mPa·s), acetic acid (CH_3_COOH, >99.5%), zinc oxide (ZnO, 99–100.5%), D-3-phenyllactic acid (PA, *M*_W_ = 166.17), glycerol (99.0%), tryptone, yeast extract, sodium chloride (NaCl), glucose (≥99.5%), and agar were purchased from Shanghai Aladdin Biochemical Technology Co., Ltd. (Shanghai, China). *Escherichia coli* (*E. coli*), *Staphylococcus aureus* (*S. aureus*), *Penicillium rubens* (*P. rubens*), and *Aspergillus niger* (*A. niger*) stains were provided by Taizhou University.

### 2.2. Fabrication of Cellulose/Chitosan/ZnO/PA (CZP) Composite Membranes

Synthesis of film liquid: A sample comprising 4 g of dried and crushed sugarcane fiber was mixed with 80 mL deionized water, and the fiber slurry was dispersed by ultrasonication (480 W, 30 min). Then, 0.25% ZnO was added and sonicated for 15 min to obtain the homogenized mixture. Next, 2 wt % chitosan/1 wt % glacial acetic acid and different amounts of PA solution (0.0, 0.3, 0.5, and 0.7%) were added to completely dissolve the mixture. The final mixture was heated and stirred continuously at 60 °C, the 0.2% glycerol was added as the plasticizer and stirred for another 20 min to obtain the film liquid. 

After passive cooling to room temperature, the film liquid was sonicated for 30 min, spread and dried at 40 °C for 24 h, resulting in the CZP composite membrane (Figure 1 and Table 1). 

Control samples: The synthesis method for the control samples was the same as that of the CZP composite membrane except that ZnO and PA were not added. The control samples were named CM-30, CM-40, and CM-50 according to the amount of membrane liquid (30, 40, and 50 g) without ZnO and PA.

### 2.3. Characterization

#### 2.3.1. Morphological Observation 

The morphology of the CZP composite membranes was observed using a scanning electron microscope (SEM, S-4800, Hitachi, Tokyo, Japan) and samples were commissioned for testing in Taizhou university.

#### 2.3.2. Fourier Transform Infrared (FTIR) Spectroscopy

The dried sample was mixed with KBr at a ratio of 1:50 and ground uniformly, after which the powder was transferred into a mold and pressed into a transparent sheet using a tablet press. The FTIR spectrum was recorded on a Nicolet 5700 instrument, Thermo Electron (Waltham, MA, USA), with 16 scans at an interval of 2 cm^−1^.

#### 2.3.3. X-ray Diffraction (XRD)

Each sample was spread evenly on a glass slide, and the XRD pattern was recorded using a D8 Advance diffractometer, Bruker (Bremen, Germany). The test voltage was 40 kV with the current 40 mA, the scanning range was 10 to 70°, and the scanning speed was 5°/min.

#### 2.3.4. Analysis of Mechanical Properties

Each neat 100 × 15 mm sample of CZP composite membrane was tested for tensile strength (TS) and elongation at breaking (EB) using a universal material testing machine at an ambient temperature of 20 ± 2 °C. Three sets of valid data were recorded for each set of measurements, whereby the maximal stress and maximum strain were recorded and averaged for analysis. The initial scale distance was set to 60 mm, and the tensile rate was 5 mm/min.

#### 2.3.5. Analysis of Hygroscopicity and Water Solubility

Samples comprising neat CZP composite membrane with a size of 40 *×* 40 mm were weighed and recorded as *W*_1_. The weighed samples were then submerged in a sufficient amount of deionized water and left for 24 h at room temperature. The samples were taken out after 24 h to remove the surface water with absorbent paper, weighed and recorded as *W*_2_. Then, the samples were dried at 40 °C, weighed and recorded as *W*_3_. Each sample was tested in triplicate, and the average value was calculated. The water absorption rate was calculated using the formula: *A = (W*_2_
*− W*_3_*)/W*_3_ × 100%.

#### 2.3.6. Water Solubility Test 

Water solubility was tested the same as hygroscopicity. Each sample was tested in triplicate. The water solubility was calculated using the formula: *Q = (W*_1_
*− W*_3_*)/W*_1_ × 100%.

### 2.4. Antimicrobial Performance of CZP Composite Membrane

In this experiment, two bacteria and two fungi (*S. aureus*, *E. coli*, *A. niger*, and *P. rubens*) were selected to verify the antimicrobial performance of CZP. The CZP composite membrane was cut into discs with a diameter of 8 mm, and the antimicrobial performance was assessed by measuring the inhibition zone surrounding each disc. The composite membrane without PA was included as a control. 100 μL of the *E. coli* or *S. aureus* cells were transferred to Luria-Bertani (LB) plate, each group of CZP composite membrane sample was placed, and cultured at 37 °C for 16 h. Similarly, 100 μL of *A. niger* or *P. rubens* spores were cultured on a Potato Dextrose Agar (PDA) plate, and each group of CZP composite membrane was applied to the plate. The *A. niger* was incubated at 37 °C for 28 h, while the *P. rubens* was at 30 °C for 72 h. The diameter of the inhibition zone was measured and averaged. The chitosan/cellulose film was as the control group to calculate the *p*-value using the Student’s *t*-test. 

## 3. Results and Discussion

### 3.1. Micromorphology of CZP Composite Membrane

To investigate the effects of PA addition on the composite membrane, electron microscopy was used to analyze the surface morphology and cross-section. As shown in Figure 2, the surface of CZP composite membranes was rough with large cellulose particles, probably due to the uneven precipitation of cellulose when forming the composite membranes [44]. The surface of CZP-50/7 was covered with dense needle-like crystals, which were consistent with the crystal structure of zinc phenyllactate [45], since the cross-section of CZP-50/0 composite membrane was flat and uniform. Compared to CZP-50/0, the cross-section of CZP-50/7 composite membrane was uneven and exhibited porous lamellar morphology with a dense structure. However, there were no obvious fractures and splits in the surface, indicating that there was good compatibility among the components.

### 3.2. Physicochemical Properties of CZP Composite Membranes

In order to analyze the chemical structure of the composite membranes, FTIR and XRD analyses were carried out. As shown in Figure 3, cellulose and chitosan have similar chemical groups and structures with similar absorption peaks. The superimposed absorption peak of chitosan at 3435.5 cm^−1^ was for the formation of hydrogen bonds between O-H and N-H, while the absorption peak of cellulose at 3434.4 cm^−1^ was only for the O-H stretching vibration [46]. The absorption peaks of cellulose and chitosan near 1637.0 cm^−1^ correspond to the C=O stretching vibration [47]. The lower absorption intensity of chitosan at 2918.9 cm^−1^ corresponded to the stretching vibrational of the methyl group on the residual sugar group, demonstrating the low crystallinity of this chitosan sample [48]. Furthermore, the peaks at 1166.0 cm^−1^ of cellulose and 1151.3 cm^−1^ of chitosan correspond to the asymmetric stretching vibration of C-O-C. The absorption peak of zinc oxide at 3434.5 cm^−1^ was ascribed to the stretching vibration of adsorbed O-H generated by the dissociation of water adsorbed on the surface of ZnO, which can be involved in the formation of hydrogen bonds, while the characteristic peak at 464.9 cm^−1^ has a sharp peak shape, indicating better FT-IR absorption [49,50,51]. The absorption peak of PA at 3446.4 cm^−1^ corresponds to its O-H stretching vibration, the peak at 1731.3 cm^−1^ corresponds to the C=O stretching vibration, and 1603.1 to 1432.3 cm^−1^ were the C=C stretching vibration in the aromatic ring.

The stretching vibration absorption peaks of O-H and N-H in cellulose and chitosan were retained after the addition of ZnO and PA, while the C-O-C asymmetric stretching vibration peak disappeared, indicating the formation of new hydrogen bonds among the components and the retention of the main framework structure of cellulose and chitosan [52]. The addition of ZnO and PA induced only a partial red shift of the peaks and a change of intensity, demonstrating that the addition of both compounds only introduced interactions between the components without changing the main structure of the composite membrane [53]. The C-O stretching vibration peaks at 1077.4 cm^−1^ in chitosan and 1059.1 cm^−1^ in cellulose were red-shifted in the composite membrane, indicating strong hydrogen bonding and good compatibility of the components in the blended system [54].

Changes in the crystal structure of the raw material during the fabrication of the CZP composite membrane were characterized by XRD. As shown in Figure 4, a sharp diffraction peak appeared at 2*θ* of 14.8°, 20.7°, 23.328°, 26.3°, and 29.8°, indicating that PA was largely crystalline, reaching 96.72%, which proved that the used PA had high purity [31]. The diffraction peaks of cellulose were respectively observed at 2*θ* = 14.4°, and 22.8°, confirming that the used material was type II cellulose [55]. Chitosan has strong rigidity of the main chain due to the presence of a large number of intra- and intermolecular hydrogen bonds, which results in a certain degree of crystallinity and density, as shown in Figure 4. Strong peaks appeared at 2*θ* values of 11.8° and 20.2°, which was consistent with the literature [56]. The diffraction peaks of ZnO at 2*θ* = 31.9°, 34.5°, 36.4°, 47.7°, 56.7° and 63.0° were detected, which was consistent with the hexagonal phase of ZnO (JCPDS 36-1451 card) [57].

Compared to pure chitosan and cellulose, the diffraction peaks of CZP composite membrane exhibited that the components interacted with each other, indicating their compatibility in the blended system. The characteristic diffraction peaks of ZnO almost disappeared in the CZP composite membrane, whereby the intensity of the diffraction peak at 37.4° was weakened obviously compared with ZnO. These results may be due to the reaction of ZnO with the acid in the mixture, which may have dissolved the crystalline structure of ZnO. However, the characteristic diffraction peaks of PA still appeared, and the weakened and narrowed overlapping peak of chitosan and cellulose was observed at 2*θ* = 22.4°, confirming that the raw material was blended and the crystalline morphology was changed. Overall, the components were compatible in the composite system.

### 3.3. Analysis of the Mechanical Performance

The results of mechanical tests for each group of specimens are shown in Table 2. The test results indicated that the CZP composite membranes maintained the maximal tensile strength and elongation, possibly because ZnO can impart a better tensile strength and toughness to composite membranes [58]. In chitosan/cellulose membranes, the tensile strength was increased significantly with the increase of added membrane solution, while the elongation was not changed. When the amount of added membrane solution reached 40 g, further addition did not lead to a commensurate increase of tensile strength. In the CZP composite membranes, the elongation was increased significantly when the amount of added membrane solution was increased to 50 g, while the tensile strength decreased at the same time. It was reported that a large amount of ZnO dispersion can result in agglomeration [59], which can potentially explain this result.

### 3.4. Hygroscopicity

The results of water absorption tests of CZP composite membranes with different composition ratios are shown in Table 3. The test results indicated that the water absorption of the composite membrane decreased when the PA concentration was increased. The change was significant when the PA concentration was increased to 0.5% (amount of membrane liquid: 30/40 g) and 0.3% (amount of membrane liquid: 50 g). This may be caused by the molecular structure and high crystallinity of PA, which results in high water resistance of the composite membranes [60].

### 3.5. Water Solubility Test

The results of the water solubility test of CZP composite membranes are shown in Table 4. The increase in the amount of liquid among other membrane components did not affect the water solubility. The addition of 0.3% PA only results in a small increase of the water solubility, while a further increase of the PA concentration led to a decrease of the water solubility.

### 3.6. Analysis of Antimicrobial Properties of CZP Composite Membranes

#### 3.6.1. Antimicrobial Activity against *S. aureus* and *E. coli*

*S. aureus* is a common Gram-positive pathogen whose control is of significant concern [61]. As shown in Figure 5, all the CZP composite membranes showed a good antibacterial effect against *S. aureus*, which was not significantly correlated with the PA concentration. Nevertheless, the antimicrobial activity was decreased when the PA concentration was low. Overall, the CZP composite membranes with 0.3% PA (CZP-30/3, CZP-40/3, and CZP-50/3) had the best effect against *S. aureus*. These results demonstrated that the antibacterial activity of the CZP composite membrane can be improved to some extent with the increase of the added composite membrane solution.

As shown in Figure 6, the inhibitory effect against *E. coli* was similar to the results obtained with *S. aureus*. With the increase of added CZP composite membrane solution, the inhibition ability was enhanced, and was weakened when the PA concentration was increased. In this test, the composite membrane with 0.3% PA showed the strongest effect against *E. coli*. Consistent with previous reports [31], *E. coli* was less sensitive to the inhibitory components in the CZP composite membrane than *S. aureus*.

#### 3.6.2. Antimicrobial Test to *Fungi*. (*A. niger* and *P. rubens*)

All CZP composite membranes exhibited good antifungal activity against *A. niger* (Figure 7). Different from the antibacterial activity against *S. aureus* and *E. coli*, the antifungal activity was improved with the increase of PA concentration in the co-mixing system with a small amount of composite membrane solution (30 g). Nevertheless, additional PA decreased the inhibitory effect of the composite membrane against *A. niger*.

The results of inhibition tests on *P. rubens* are shown in Figure 8. Compare with *A. niger*, *P. rubens* was less sensitive, and only 0.7% PA or the addition of more than 40 g membrane liquid resulted in growth inhibition. Furthermore, membranes with only ZnO or chitosan had a weak effect on *P. rubens*. With the increase of PA concentration, the inhibition ability was enhanced, and the membrane with 0.7% PA had the best effect against *P. rubens*.

## 4. Conclusions

In this study, chitosan and cellulose were used as film-forming materials, glycerol was used as a plasticizer, while the inorganic agent ZnO and natural compound PA were used as the main antimicrobial components to prepare composite membranes. FTIR, XRD, and SEM analyses demonstrated the interactions between the components of composite membranes and confirmed their good blending compatibility. The addition of ZnO enhanced the mechanical properties of composite membranes, while the combination of ZnO and PA reduced their water absorption and swelling properties. The antimicrobial inhibition zones surrounding membrane discs on agar plates indicated that the addition of PA improved the antimicrobial activity of the composite membrane, with good effects against *S. aureus*, *E. coli*, *A. niger* and *P. rubens*. The CZP composite membrane had the best effect against *S. aureus*, *E. coli* and *A. niger* with the addition of 0.3% PA, and the best effect against *P. rubens* with the addition of 0.7% PA. These adjustable CZP composite membranes have great potential for use in biodegradable antimicrobial packaging applications.

## Figures and Tables

**Figure 1 membranes-12-00239-f001:**
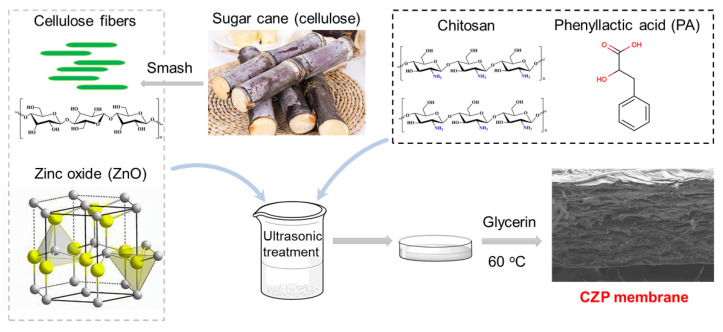
Schematic diagram of the preparation of the CZP composite biodegradable antimicrobial membrane doped with organic/inorganic antimicrobial agents.

**Figure 2 membranes-12-00239-f002:**
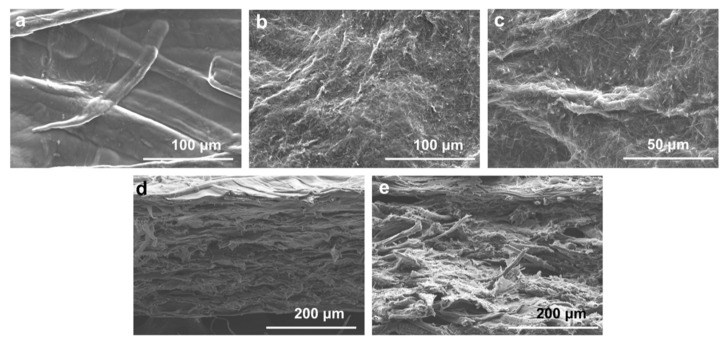
SEM images of composite membranes. (**a**) SEM of CZP-50/0; (**b**,**c**) The membrane surface of CZP-50/7 composite; (**d**) The cross section of CZP-50/0; (**e**) The cross section of CZP-50/7.

**Figure 3 membranes-12-00239-f003:**
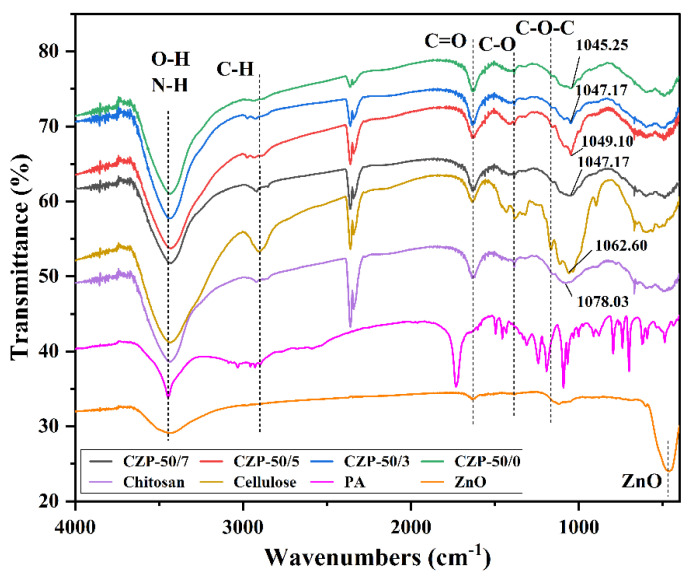
FTIR spectra of pure cellulose, pure chitosan, PA, ZnO and CZP composite membranes with different composition ratios.

**Figure 4 membranes-12-00239-f004:**
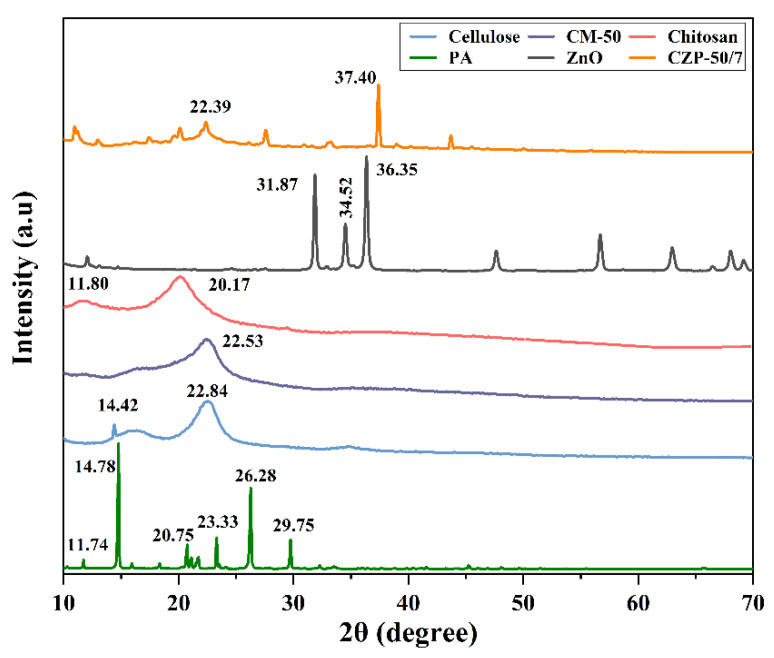
XRD patterns of PA, pure cellulose, pure chitosan, ZnO and CZP composite membranes with different composition ratios.

**Figure 5 membranes-12-00239-f005:**
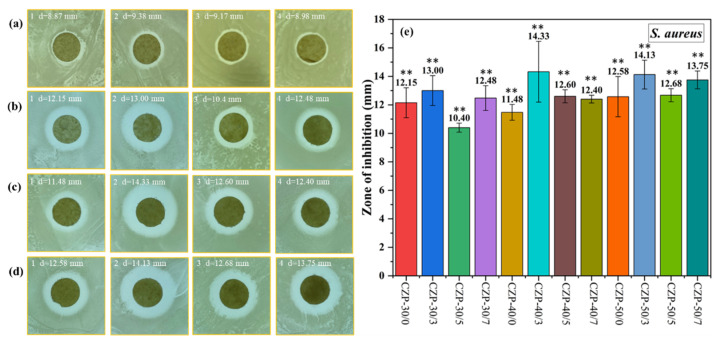
Digital photos of the composite membranes without ZnO and PA ((**a**) 1–4) and CZP composite membranes on plates with bacterial lawns of *S. aureus*. ((**b**) 1–4) CZP-30/0, CZP-30/3, CZP-30/5, and CZP-30/7. ((**c**) 1–4) CZP-40/0, CZP-40/3, CZP-40/5, and CZP-40/7. ((**d**) 1–4) CZP-50/0, CZP-50/3, CZP-50/5, and CZP-50/7. (**e**) Inhibition zones of the CZP composite membranes against *S. aureus* (* *p* < 0.05, ** *p* < 0.01).

**Figure 6 membranes-12-00239-f006:**
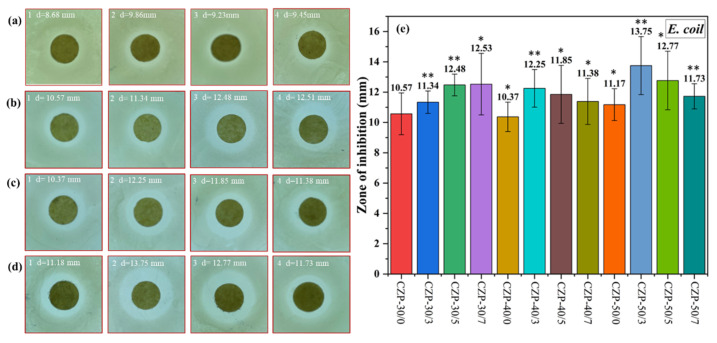
Digital photos of the composite membranes without ZnO and PA ((**a**) 1–4) and CZP composite membranes on plates with bacterial lawns of *E. coli*. ((**b**) 1–4) CZP-30/0, CZP-30/3, CZP-30/5, and CZP-30/7. ((**c**) 1–4) CZP-40/0, CZP-40/3, CZP-40/5, and CZP-40/7. ((**d**) 1–4) CZP-50/0, CZP-50/3, CZP-50/5, and CZP-50/7. (**e**) Inhibition zones of the CZP composite membranes against to *E. coli* (* *p* < 0.05, ** *p* < 0.01).

**Figure 7 membranes-12-00239-f007:**
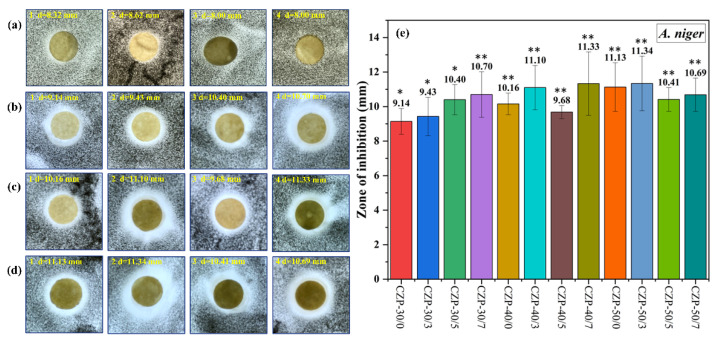
Digital photos of composite membranes without ZnO and PA ((**a**) 1–4) and CZP composite membranes on plates with bacterial lawns of *A. niger*. ((**b**) 1–4) CZP-30/0, CZP-30/3, CZP-30/5, and CZP-30/7. ((**c**) 1–4) CZP-40/0, CZP-40/3, CZP-40/5, and CZP-40/7. ((**d**) 1–4) CZP-50/0, CZP-50/3, CZP-50/5, and CZP-50/7. (**e**) Inhibition zones of the CZP composite membranes against to *A. niger* (* *p* < 0.05, ** *p* < 0.01).

**Figure 8 membranes-12-00239-f008:**
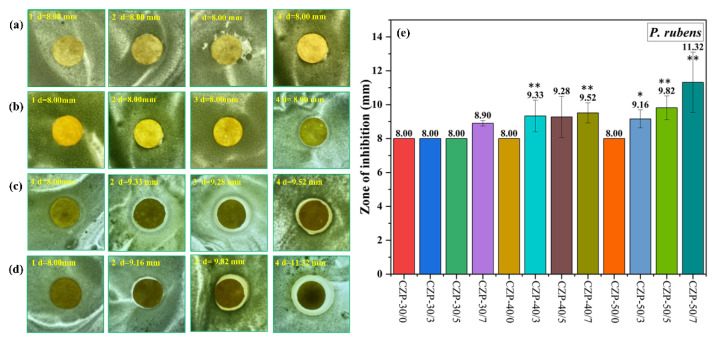
Digital photos of composite membranes without ZnO and PA ((**a**) 1–4) and the CZP composite membranes on plates with bacterial lawns of *P. rubens*. ((**b**) 1–4) CZP-30/0, CZP-30/3, CZP-30/5, and CZP-30/7. ((**c**) 1–4) CZP-40/0, CZP-40/3, CZP-40/5, and CZP-40/7. ((**d**) 1–4) CZP-50/0, CZP-50/3, CZP-50/5, and CZP-50/7. (**e**) Antimicrobial data of the CZP composite membranes against to *P. rubens* (* *p* < 0.05, ** *p* < 0.01).

**Table 1 membranes-12-00239-t001:** The sample names based on the different component ratios.

Amount of Film Liquid Added(g)	PA Concentration (%)			
	0	0.3	0.5	0.7
30	CZP-30/0	CZP-30/3	CZP-30/5	CZP-30/7
40	CZP-40/0	CZP-40/3	CZP-40/5	CZP-40/7
50	CZP-50/0	CZP-50/3	CZP-50/5	CZP-50/7

**Table 2 membranes-12-00239-t002:** Mechanical performance of cellulose chitosan membranes and CZP composite membranes.

Sample	Cellulose/Chitosan Membranes		Sample	CZP Composite Membranes	
	Tensile Strength /MPa	Elongation /%		Tensile Strength /MPa	Elongation /%
CM-30	5.77 ± 0.025	11.51 ± 0.054	CZP-30/0	9.48 ± 0.047	12.38 ± 0.067
CM-40	8.45 ± 0.042	10.59 ± 0.052	CZP-40/0	10.69 ± 0.051	12.06 ± 0.062
CM-50	8.46 ± 0.041	11.11 ± 0.049	CZP-50/0	7.72 ± 0.036	16.63 ± 0.062

**Table 3 membranes-12-00239-t003:** The hygroscopicity (%) of CZP composite membranes.

Amount of Film Liquid Added(g)	PA Concentration (%)			
	0	0.3	0.5	0.7
30	7.11 ± 0.095	7.63 ± 0.057	3.60 ± 0.031	2.74 ± 0.010
40	7.21 ± 0.011	8.90 ± 0.197	3.97 ± 0.025	4.21 ± 0.015
50	7.40 ± 0.041	4.16 ± 0.015	4.33 ± 0.049	2.42 ± 0.073

**Table 4 membranes-12-00239-t004:** The water solubility (%) of CZP composite membranes.

Amount of Film Liquid Added(g)	PA Concentration (%)			
	0	0.3	0.5	0.7
30	43.06 ± 0.050	46.81 ± 0.042	42.61 ± 0.039	42.43 ± 0.007
40	41.23 ± 0.004	50.35 ± 0.031	41.98 ± 0.005	44.34 ± 0.003
50	41.22 ± 0.018	41.27 ± 0.010	42.24 ± 0.014	39.21 ± 0.001

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
