# Peer review of "Enhanced Antimicrobial Cellulose/Chitosan/ZnO Biodegradable Composite Membrane"

_membranes, 2022, doi:10.3390/membranes12020239_

Round 1

Reviewer 1 Report

The authors have amended some of the problems with the original manuscript, namely the insufficient description of the sample preparation process. However, many problems persist: Fig. 3 has obviously not been improved as suggested in the previous review, the characteristic spectra of cellulose, chitosan and the CZP samples are still indiscernible due to the improper range of the y-axis. The only present spectrum, where the characteristic IR bands can be somewhat recognized is the spectrum of PA, however, it is still not very clear due to the low resolution of the image. ZnO does not show significant bands in IR spectrum, and it can be omitted from the image. The text describing the spectra was improved, however it is still far from satisfactory. In Fig. 4 the XRD spectrum of PA is still strangely excluded from the image with the spectra of the other samples, the bands in the PA XRD spectrum should be marked in the image as well. The biggest problem I have with the manuscript are the antimicrobial test results. Now that the error bars were added to the results, it is really evident the addition of PA does not have statistically significant influence on the inhibition zone in the CZP samples. I am also unsure, why samples with different thickness are compared in these tests, the film thickness clearly should not have much influence on the antimicrobial effect of the samples if the amount of the material remains the same. Another problem is that the control (e.g. the CM samples) is still missing and it can not be concluded the antimicrobial effect is due to the presence of PA or ZnO in the samples. It is known that chitosan by itself can act as an antibacterial agent, the antimicrobial activity can also be influenced by residual acetic acid used during preparation of the samples.

These are fundamental issues with the scientific validity of the manuscript (besides numerous problems with English grammar throughout the manuscript), which have not been addressed during the revision of the paper. Therefore, I have to recommend the Membranes journal editorial office to reject the manuscript for publication.

Reviewer 2 Report

Thank you very much about your revisions to improve the manuscript.

I suggest it is suitable for publication after correct this point.

  • In Figure 1. Schematic diagram of the preparation of CZP composite degradable antimicrobial membrane modified by the addition of organic/inorganic antimicrobial agents, you showed that the addition of Chitosan before ZnO and PA, while in 2. Fabrication of cellulose/chitosan/ZnO/PA (CZP) composite membranes section, you showed that ZnO first then Chitosan added, why?

Reviewer 3 Report

The authors did not rewrite the introduction and did not use the suggested references in the article.      

Round 2

Reviewer 1 Report

The authors have significantly improved the manuscript quality in this revision. The previously stated problems were alleviated and the manuscript is now in form, where it can be published. I would advise the authors to also consider adding the antimicrobial results of the control samples either in the manuscript or in supplementary data.

Reviewer 3 Report

It is acceptable.
